# Gait and Dual-Task Performance in Older Adults with Suspected Cognitive Impairment: Effects of an 8-Week Exercise Program

**DOI:** 10.3390/healthcare13243190

**Published:** 2025-12-05

**Authors:** João Galrinho, Marco Batista, Marta Gonçalves-Montera, Orlando Fernandes, Ana Rita Matias

**Affiliations:** 1Comprehensive Health Research Center (CHRC), Department of Sports and Health, School of Health and Human Development, University of Évora, 7006-071 Évora, Portugal; orlandoj@uevora.pt (O.F.); armatias@uevora.pt (A.R.M.); 2Sport Physical Activity and Health Research & Innovation Centre, SPRINT Polytechnic University of Castelo Branco, 6000-084 Castelo Branco, Portugal; marco.batista@ipcb.pt; 3Faculty of Psychology, University of Lisbon, 1649-013 Lisbon, Portugal; martagoncalves@psicologia.ulisboa.pt

**Keywords:** Timed Up and Go, dual-task, gait, cognitive impairment, institutionalized aging, exercise intervention

## Abstract

**Highlights:**

Older adults with suspected cognitive impairment showed significantly poorer single- and dual-task mobility at baseline, confirming that cognitive decline substantially increases dual-task cost and functional vulnerability.An 8-week multicomponent exercise program produced large, meaningful improvements in TUG and TUG-DT performance in cognitively impaired participants, demonstrating the effectiveness of short-term, cognitively enriched training in enhancing mobility.

**What are the main findings?**
Older adults with suspected cognitive impairment performed worse than those without impairment on TUG and TUG-DT at baseline; the impaired group improved substantially after an 8-week multicomponent program.Simple TUG showed the largest responsiveness post-intervention; TUG-DT captured cognition–mobility demands.

**What are the implications of the main findings?**
TUG measures are sensitive to cognitive status and change, supporting routine functional screening in institutional settings.Personalized multicomponent exercise may enhance mobility in cognitively impaired older adults.

**Abstract:**

**Background/Objectives:** Gait performance in aging relies heavily on cognitive resources, yet the extent to which short-term interventions can mitigate dual-task costs in institutionalized populations remains understudied. This study aimed to compare single and dual-task gait performance between older adults with and without suspected cognitive impairment and to evaluate the effects of an 8-week multicomponent exercise program on functional mobility. **Methods:** Institutionalized older adults (*n* = 42) were stratified into two groups: suspected cognitive impairment (*n* = 26) and no suspected impairment (*n* = 16), based on MMSE and Clock Drawing Test screening. Participants performed the Timed Up and Go (TUG) and Dual-Task TUG (TUG-DT) at baseline and post-intervention. **Results:** At baseline, the suspected impairment group exhibited significantly poorer performance on both tests (*p* < 0.001) compared to the non-impaired group. Following the 8-week intervention, the suspected impairment group demonstrated large, significant improvements in both TUG (*r* = −0.73) and TUG-DT (*r* = −0.59), whereas the non-impaired group remained stable. Notably, while the single-task TUG showed the greatest responsiveness to the exercise program, the TUG-DT continued to reveal a significant cognitive-motor load. **Conclusions:** Multicomponent exercise effectively enhances functional mobility in cognitively vulnerable older adults, reversing declines in both single and dual-task conditions. **Significance**: These findings support the implementation of dual-task screening to unmask latent functional deficits and validate the use of accessible, short-term multicomponent exercise programs as a vital strategy to preserve autonomy in institutionalized older adults.

## 1. Introduction

Population aging represents a pressing global challenge accompanied by profound transformations in functional capacity [1]. Gait and mobility are essential for independence [2,3], yet age-related alterations compromise autonomy and increase the risk of adverse outcomes [4]. Walking speed typically remains stable until approximately 70 years of age before declining [5], often accompanied by increased double-support time to enhance stability [6,7,8]. These alterations result from multifactorial mechanisms, including the loss of gastrocnemius power affecting the push-off phase [9,10], and postural adaptations such as anterior pelvic tilt associated with muscle weakness and adiposity [11,12]. Sarcopenia [13,14], joint pathologies [15,16], and neural declines [17,18,19,20] further compound these deficits.

Within this framework, cognitive impairment has emerged as a critical determinant of gait performance. Encompassing a spectrum from mild cognitive impairment (MCI) to major neurocognitive disorders [21,22], cognitive decline is associated with slower gait and increased variability [23,24]. Gait requires continuous executive control [25,26]; when these processes are impaired, individuals struggle to allocate attentional resources [27,28]. This relationship is bidirectional: mobility limitations can exacerbate cognitive deficits, while cognitive impairment further compromises gait [29].

To assess this interaction, the Timed Up and Go (TUG) test is widely adopted for its predictive value [30,31,32] and sensitivity to cognitive changes [23,33]. The Dual-Task TUG (TUG-DT) provides deeper insight into cognitive-motor interference [34,35]. Quantifying dual-task cost is clinically valuable [36], as performance deterioration is accentuated in individuals with sarcopenia or cognitive impairment [37]. Consequently, gait assessment under dual-task paradigms is a sensitive method for identifying early dysfunction [38], with dual-task cost serving as a predictor of falls and executive decline [24,39,40,41].

Interventions addressing both domains are therefore crucial. Multicomponent programs combining physical and cognitive stimulation have yielded superior improvements compared to physical training alone [42]. Specifically, exercise incorporating cognitive challenges (motor-cognitive training) enhances both gait and executive function [43,44], supporting the attentional resource-sharing theory [45]. However, the extent to which functional mobility can be preserved in institutionalized populations through short-term interventions remains a key area of investigation. Therefore, this study aims to compare TUG performance under dual-task conditions between older adults with and without suspected cognitive impairment and to evaluate the efficacy of an 8-week multicomponent exercise program.

In this context, the present study defines its research framework using the PECO structure. The target Population (P) consists of institutionalized older adults, stratified into two groups: those with suspected cognitive impairment and those without suspected cognitive impairment. The Exposure (E) analyzed is an 8-week multicomponent exercise program designed to integrate physical and cognitive stimuli. The Comparator (C) is established at two levels: between-group comparisons (impairment vs. no impairment) and intra-subject comparisons between test conditions (single-task vs. dual-task). The Outcomes (O) of interest focus on gait differences and functional mobility, assessed through performance on the Timed Up and Go (TUG) and Dual-Task TUG (TUG-DT), as well as the response to the intervention. By applying both the standard and dual-task versions of the TUG test, we examine how cognitive load influences motor execution and how the intervention impacts functional mobility. This design allows for a nuanced understanding of whether improvements in single-task performance are paralleled, or limited by gains under dual-task conditions. Specifically, it investigates whether the TUG-DT remains a more demanding condition for individuals with cognitive impairment, even after a structured intervention, thereby revealing persistent challenges in attention sharing and executive control. The study ultimately seeks to clarify the degree to which functional mobility can be achieved and maintained in cognitively vulnerable populations through short-term, targeted programs. By integrating both single- and dual-task assessments, this research aims to advance the use of gait analysis as a complementary tool for the early detection of cognitive and functional decline. Its findings are expected to contribute to the refinement of clinical assessment protocols and the development of multidimensional intervention strategies that preserve autonomy, reduce fall risk, and improve quality of life among older adults. In doing so, it aligns with current priorities in geriatric healthcare and neurorehabilitation that emphasize the interdependence of physical and cognitive health across the aging continuum.

Therefore, based on the theoretical assumptions regarding the interdependence between cognition and functional mobility, the specific objectives of this study were to compare TUG performance under dual-task conditions between older adults with and without suspected cognitive impairment, and to analyze the dual-task cost at baseline to determine if cognitive load induces more pronounced motor degradation in the impaired group. Furthermore, we aimed to evaluate the efficacy of the 8-week multicomponent exercise program in improving functional performance across both single- and dual-task conditions, while comparing the sensitivity of the single-task versus dual-task TUG in detecting post-intervention changes. Finally, the study sought to verify whether the dual-task condition continues to represent a significantly higher functional demand than the single-task condition after the intervention, thereby indicating persistent limitations in cognitive-motor integration within the suspected impairment group.

The hypotheses of this study assume that older adults with suspected cognitive impairment exhibit poorer functional mobility compared with those without impairment, and that this difference becomes more pronounced when a simultaneous cognitive task is introduced, reflecting greater vulnerability to dual task cost. It is further anticipated that the eight-week multicomponent exercise program will improve functional mobility, particularly in the group with suspected cognitive impairment. Nevertheless, dual task performance is expected to remain more demanding than single task performance after the intervention, suggesting that cognitive–motor interference does not fully disappear. Finally, it is anticipated that the single task TUG will be more sensitive to training-induced improvements than the dual task TUG, indicating that basic motor recovery tends to occur before the optimization of the more complex mechanisms of attentional sharing.

## 2. Materials and Methods

### 2.1. Ethics Approval

The Évora University research ethics committee approved all the procedures. In addition, participants provided written informed consent in accordance with the Helsinki Declaration before participating in this study. All participants gave their verbal consent.

### 2.2. Participants

This study followed a quasi-experimental, longitudinal design using convenience sampling. Participants were recruited from three institutional settings and stratified into groups based on cognitive screening results, resulting in numerically unbalanced groups due to the natural prevalence of cognitive decline in this population

The study included institutionalized older adults with and without suspected cognitive impairment. In the first assessment, the sample comprised 58 participants, divided into two groups: one with suspected cognitive impairment (*n* = 38) and another without (*n* = 20). The mean age of participants with suspected cognitive impairment was 86.05 years (SD = 6.26), ranging from 62 to 99 years. In the group without suspected cognitive impairment, the mean age was 84.35 years (SD = 7.78), with ages ranging from 67 to 99 years.

Regarding gender distribution at baseline, among participants with suspected cognitive impairment, 14 (36.8%) were male and 24 (63.2%) were female. In the group without suspected cognitive impairment, 5 participants (25%) were male and 15 (75%) were female.

For the second assessment, only participants who remained institutionalized and available for follow-up were included. Accordingly, 26 individuals (68.4%) from the group with suspected cognitive impairment were retained in the second phase. The remaining individuals were excluded due to voluntary withdrawal (*n* = 4; 10.5%), change in residence (*n* = 2; 5.3%), return to home environment (*n* = 1; 2.6%), and death (*n* = 5; 13.2%). In the group without suspected cognitive impairment, 16 participants (80%) continued under investigation, while 3 (15%) returned to their homes and 1 (5%) passed away.

Thus, the second assessment sample comprised 42 participants, divided into two groups: one with suspected cognitive impairment (*n* = 26) and another without (*n* = 16). The mean age of the group with suspected cognitive impairment was 86.08 years (SD = 7.11), ranging from 62 to 99 years. In the group without suspected cognitive impairment, the mean age was 85.94 years (SD = 7.68), ranging from 67 to 99 years.

About gender distribution in the second assessment, 9 participants (34.6%) in the group with suspected cognitive impairment were male, and 17 (65.4%) were female. In the group without suspected cognitive impairment, 4 participants (25%) were male and 12 (75%) were female.

Inclusion criteria for individuals with suspected cognitive impairment were:(i)age over 60 years;(ii)cognitive impairment confirmed by medical information;(iii)a Mini-Mental State Examination (MMSE) score adjusted for education levels according to Portuguese normative data: ≤15 for illiterate individuals; ≤22 for those with 1 to 11 years of schooling; and ≤27 for those with >11 years of schooling [45];(iv)a score of 6 or lower on the Clock Drawing Test.

For individuals without suspected cognitive impairment, the inclusion criteria were as follows:

(i) documentation in the clinical record indicating that institutionalization was not due to cognitive decline, but rather to other reasons such as rehabilitation, accompanying an institutionalized spouse, or social isolation; (ii) a Mini-Mental State Examination (MMSE) score above the cut-off thresholds adjusted for education levels [46]: >15 for illiterate individuals, >22 for individuals with 1 to 11 years of formal education, and >27 for individuals with more than 11 years of formal education; (iii) a score higher than 6 on the Clock Drawing Test.

Exclusion criteria for both groups were established to minimize confounding factors that could independently affect gait or cognitive testing performance. Participants were excluded if they presented: (i) History of neurological or psychiatric disorders (e.g., stroke, Parkinson’s disease, Huntington’s disease, major depression, or psychosis) unrelated to the aging process, which could bias motor or cognitive outcomes; (ii) Severe sensory or motor impairments (e.g., uncorrected visual deficits or upper limb paralysis) that would interfere with the protocol execution or written data collection; (iii) Native language other than Portuguese, to ensure the validity of the verbal cognitive screening tools.

### 2.3. Instruments and Procedures

The neuropsychological assessment was conducted by a Psychology undergraduate and the principal investigator, under the supervision of a professional with advanced specialization in Neuropsychology. To identify suspected cognitive impairment, the following instruments were used: (i) The Mini-Mental State Examination (MMSE), widely used to assess global cognitive function in cognitive screening. It consists of 30 items covering six cognitive domains [47]; (ii) the Clock Drawing Test, which has been broadly applied in neurological, psychiatric and psychological evaluations. It is increasingly common as a screening tool for cognitive changes associated with aging [48]. The first study associating this test with the identification of older adults with cognitive deficits was conducted in 1986 [49]. This test evaluates multiple cognitive functions and is comparable to the MMSE [50]; (iii) Cognitive Impairment Confirmed by Medical Information, including clinical records, neuropsychological assessments and observational data from daily routines.

Psychomotor assessment was conducted exclusively by an Exercise Physiologist and a Psychomotor Therapist. The TUG test was used, both in its single-task and dual-task versions. The TUG is a widely validated measure of functional mobility in older adults [30,34,51], although some authors note limitations in its sensitivity for healthy, autonomous older adults, suggesting it is more appropriate for individuals with functional deficits [52]. The TUG test reflects not only motor ability but also cognitive factors [53].

The TUG Test was administered following a preliminary trial for familiarization. Participants were seated in an armchair (seat height 44–46 cm). Upon the starting signal, the participant stood up, walked 3 m, turned, returned, and sat back down. The task was performed at the individual’s usual walking speed, wearing regular footwear and using a walking aid if necessary. According to Shumway-Cook, Brauer, and Woollacott [34], execution times of 13.5 s or greater indicate a high risk of falls in older adults [34].

For the dual-task TUG, participants followed the same procedure but were asked to name animals aloud while walking, adding a cognitive load [54]. Dual-task additions typically increase the time required to complete the TUG, with cognitive tasks having a greater impact on performance than manual tasks [54,55].

The multicomponent intervention program had a total duration of 8 weeks, with 45 min sessions held three times a week on non-consecutive days, totaling 24 sessions. The planning and implementation of the program were conducted by a certified Exercise Physiologist with over 10 years of professional experience in geriatrics, assisted by a Psychomotor Therapist. To ensure safety and the quality of technical supervision, as well as to allow for individualized feedback and the correction of movement patterns, participants were organized into small groups of 4 to 6 individuals, maintaining a supervision ratio of approximately 1:5.

The training protocol followed the principles of progressive overload and specificity, structured into three distinct phases: (1) Adaptation Phase (weeks 1–2), focused on familiarization with exercises, technique acquisition, and safety; (2) Development Phase (weeks 3–6), characterized by a progressive increase in the volume and complexity of motor tasks; and (3) Consolidation Phase (weeks 7–8), where intensity and cognitive demand were maximized. Each session followed a standardized structure, composed of 10 min of warm-up (joint mobility and cardiorespiratory activation), 25 to 30 min of the main phase (multicomponent exercises), and 5 to 10 min of cool-down (static stretching and breathing exercises).

The core component of the program integrated muscular resistance, static and dynamic balance, and gait training exercises, performed in a circuit or stations. A distinctive element of this intervention was the systematic incorporation of simultaneous cognitive challenges (dual-task) during motor execution, particularly starting from the development phase. These cognitive tasks included: (a) verbal fluency (e.g., naming animals, fruits, or cities while navigating an obstacle course); (b) memory tasks (e.g., memorizing and reproducing sequences of colors or movements); and (c) simple mental calculation (e.g., counting backwards or performing basic arithmetic operations during balance exercises). Music was used as a facilitating element to stimulate rhythm and auditory-motor coordination.

Attendance was recorded at every session to monitor program adherence. To be included in the final analysis, participants were required to meet a minimum attendance rate of 75% (at least 18 out of 24 sessions). Sessions missed due to occasional health issues or personal reasons were not individually rescheduled; however, the technical team ensured closer monitoring and support in the subsequent session to mitigate any loss of progression. The average adherence rate recorded in the study was high, demonstrating the acceptance and feasibility of the protocol within this institutionalized population.

### 2.4. Statistical Analysis

The data were analyzed using IBM SPSS Statistics, version 30.0. Descriptive statistics (mean, standard deviation, and interquartile range) were calculated to characterize the sample. Normality was assessed using the Shapiro–Wilk test. Given the non-normal distribution and the sample size, nonparametric tests were used.

Intra group comparisons (pre versus post intervention) were performed using the Wilcoxon test. To directly analyze cognitive cost, the differences between single task (TUG) and dual task (TUG-DT) performance were assessed using the Wilcoxon test within each group. Between group comparisons (Impairment vs. No Impairment) were conducted using the Mann–Whitney U test.

To reduce the risk of Type I error associated with multiple comparisons, a Bonferroni correction was applied to the significance threshold (adjusted alpha). Effect sizes were calculated based on Rosenthal’s r classification (r < 0.1 negligible, 0.1–0.3 small, 0.3–0.5 medium, ≥0.5 large) [56]. All analyses adopted a 95 percent confidence level.

## 3. Results

This section presents the main findings obtained in our study. According to the results shown in Table 1, in the first assessment, the group with suspected cognitive impairment exhibited a significantly higher mean time in the Timed Up and Go (TUG) test compared to the group without suspected cognitive impairment (21.30 ± 6.07 s vs. 11.14 ± 1.31 s; *p* < 0.001). This difference was even more pronounced under the dual-task condition, with the group with cognitive impairment registering a mean of 27.24 ± 7.66 s, whereas the group without cognitive impairment showed 14.28 ± 1.49 s (*p* < 0.001).

In the second assessment, conducted after the intervention, the group with suspected cognitive impairment demonstrated significant performance improvement, with a reduction in average times on the TUG (17.84 ± 3.12 s) and the TUG under dual-task conditions (24.53 ± 5.86 s). In contrast, the group without suspected cognitive impairment showed a slight variation in average times, with an increase to 12.38 ± 1.30 s on the TUG and 14.66 ± 2.20 s under the dual-task condition. However, these differences are not clinically relevant and may be attributed to natural fluctuations, individual variability or contextual factors, such as fatigue or motivation. It is essential to highlight that, despite this variation, the functional performance of the group without suspected cognitive impairment remained significantly superior to that of the group with suspected cognitive impairment in both tests (*p* < 0.001).

Regarding the specific impact of the cognitive load, a direct comparison using the Wilcoxon Signed-Rank Test revealed that TUG-DT execution times were significantly higher than single-task TUG times in both groups at baseline (*p* < 0.001). This statistically confirms the presence of a ‘dual-task cost’ across the sample. Following the intervention, despite the motor improvements observed, the suspected impairment group continued to exhibit significantly slower times on the TUG-DT compared to the single-task TUG (*p* < 0.001), indicating that the cognitive task remained a significant constraint on mobility.

According to the results presented in Table 2, in the intra-group analysis (pre- and post-intervention), the group with suspected cognitive impairment showed statistically significant improvements in the TUG Test (TUG; *p* < 0.001) and in the TUG under dual-task conditions (TUG Dt; *p* = 0.003). These results demonstrate a positive effect of the intervention on functional mobility and the ability to perform simultaneous tasks, suggesting gains in both motor and cognitive domains.

In the group without suspected cognitive impairment, a statistically significant difference was observed in the TUG (*p* = 0.026), reflected by a slight increase in execution time. Although statistically significant, this difference is likely to have no immediate clinical implications. In the TUG-DT, no statistically significant changes were observed (*p* = 0.44), indicating stability in functional performance in this type of task.

The effect size analysis, calculated through Rosenthal’s *r* coefficient in the Wilcoxon test, revealed that the intervention applied to the group with suspected cognitive deficit had a large impact on both analyzed variables. In the TUG test, the effect size was large (*r* = −0.73; *p* < 0.001), indicating a substantial improvement in functional mobility. Similarly, in the TUG-DT, the effect size was large (*r* = −0.59; *p* = 0.003), indicating significant gains in the ability to perform simultaneous tasks under additional cognitive load.

In the group without suspected cognitive impairment, the TUG test also showed a large effect size (*r* = −0.56; *p* = 0.026). In the dual-task TUG, the effect size was small (*r* = −0.19; *p* = 0.44), with no statistically significant change, reinforcing the stability of functional performance in this group.

## 4. Discussion

The present study aimed to analyze the interaction between cognitive impairment and functional mobility in institutionalized older adults, as well as to evaluate the impact of a multicomponent exercise program. The discussion is structured according to the study’s integrated hypotheses, interpreting the findings in light of current literature.

### 4.1. Baseline Differences and Dual-Task Cost (Hypothesis 1)

The results confirm the first hypothesis, demonstrating that older adults with suspected cognitive impairment exhibit significantly lower functional mobility compared to those without suspected impairment. At baseline, the suspected impairment group took nearly twice as long to complete the single-task TUG compared to the non-impaired group. As hypothesized, this difference was accentuated by the introduction of a simultaneous cognitive task. In the dual-task TUG (TUG-DT), the suspected impairment group recorded a substantial increase in execution time, validating the premise that dual-task cost is a distinctive marker of functional decline associated with cognitive impairment. This aligns with the attentional resource-sharing theory [45], suggesting that in the presence of cognitive deficits, the competition for neural resources between gait and the cognitive task compromises motor stability.

### 4.2. Impact of Intervention and Persistence of Cognitive Load (Hypothesis 2)

The hypothesis that the 8-week exercise program would improve functional mobility in the group with suspected cognitive impairment was confirmed. Post-intervention, this group demonstrated significant reductions in execution times for both single and dual-task conditions. These findings corroborate previous evidence that multicomponent exercise can promote functional gains even in cognitively vulnerable populations [42,57]. However, consistent with our prediction regarding the persistence of interference, the dual-task condition remained significantly more demanding than the single-task condition after the intervention. This suggests that while global functional capacity (speed, balance) improved, the cognitive interference did not disappear entirely. It indicates that 8 weeks may be sufficient for neuromuscular adaptations but perhaps insufficient to fully automate the management of attentional resources required for dual-tasking, suggesting a need for longer interventions to consolidate neurocognitive efficiency.

### 4.3. Sensitivity and Responsiveness of Tests (Hypothesis 3)

Contrary to some literature suggesting TUG-DT as the most sensitive marker for change, our results supported the hypothesis that the single-task TUG would be more responsive to this specific short-term intervention (showing a larger effect size). This apparent contradiction regarding sensitivity requires careful interpretation. While TUG-DT is widely recognized as superior for diagnostic screening and fall risk prediction [40]—a finding supported by our baseline data—the single-task TUG appears more sensitive to short-term motor recovery. A plausible explanation is that the recovery of basic motor parameters (speed, cadence) precedes the optimization of complex executive functions. Additionally, TUG-DT performance involves higher inter-individual variability due to anxiety or task-prioritization strategies, which may statistically dilute the magnitude of improvement compared to the simpler task.

### 4.4. Ceiling Effects and Alternative Hypotheses in the Non-Impaired Group

In the group without suspected cognitive impairment, performance remained stable, consistent with a ceiling effect [58], as these participants already exhibited high baseline functionality. However, alternative hypotheses must be considered to explain this stagnation. First, the stimulus intensity may have been insufficient for this more robust group, highlighting the need for personalized overload principles. Second, contextual factors such as lower motivation or boredom with repetitive simple testing could have masked potential minor gains. This reinforces the importance of personalizing interventions [59,60], as standardized models may fail to challenge healthier older adults sufficiently.

### 4.5. Practical Applications

The study validates the use of TUG and TUG-DT as accessible, low-cost screening tools in institutional settings [30,32]. The rapid application of these tests allows for early detection of decline, facilitating individualized program adjustments. Furthermore, the significant improvements in the impaired group suggest that implementing accessible, playful, and multicomponent exercise routines in nursing homes is a viable strategy to preserve autonomy and potentially reduce the public health costs associated with falls [33,38].

### 4.6. Limitations

Several limitations must be acknowledged. First, the study utilized a quasi-experimental design without randomization or an active control group, which limits the ability to rule out placebo effects. Second, although recruited from three different institutions, the sample sizes were unbalanced, and a significant dropout rate (31.5%) was observed in the suspected impairment group at follow-up. This attrition is reflective of the high vulnerability and advanced age of the institutionalized population (mean age ~86 years) but may impact generalizability. Furthermore, while the TUG-DT assessed the temporal cost, the performance on the secondary cognitive task (e.g., number of words) was not quantified. Consequently, we could not calculate the ‘cognitive cost’ or assess task-prioritization strategies. Future studies should record bidirectional interference. Finally, the wide age range (62–99 years) and lack of detailed sociodemographic stratification (e.g., rural vs. urban) are limitations that future research with larger samples should address to better understand intergenerational differences in dual-task adaptation.

## 5. Conclusions

This study confirms that TUG-DT is a sensitive instrument for differentiating cognitive profiles in institutionalized older adults, revealing a distinct vulnerability to cognitive-motor interference in those with suspected impairment. A central contribution of this research is the finding that, while an 8-week multicomponent exercise program effectively restores basic functional mobility (best captured by the single-task TUG), the underlying challenges in executive control taxed by the TUG-DT persist. This suggests that basic motor recovery may precede the optimization of complex attentional sharing mechanisms, indicating that longer or more cognitively targeted interventions may be required to fully resolve dual-task deficits.

Clinically, these findings reinforce two key implications: (1) the utility of the TUG-DT as an essential screening tool for early detection of cognitive-motor deficits; and (2) the urgent need to include adapted exercise programs in institutional settings to preserve autonomy. Future investigations should prioritize randomized designs with longer follow-ups to assess the consolidation of these neurocognitive gains.

## Figures and Tables

**Table 1 healthcare-13-03190-t001:** TUG Test Results in 1st and 2nd Assessment (Pre- and Post-Intervention) in Older Adults with and without Suspected Cognitive Impairment.

Assessment	With Suspected Cognitive Deficit	No Suspected Cognitive Deficit	*p*-Value
1st Assessment—TUG	(*n* = 38) 21.30 (±6.07)	(*n* = 20) 11.14 (±1.31)	<0.001
1st Assessment—TUG-DT	(*n* = 38) 27.24 (±7.66)	(*n* = 20) 14.28 (±1.49)	<0.001
2nd Assessment—TUG	(*n* = 26) 17.84 (±3.12)	(*n* = 16) 12.38 (±1.30)	<0.001
2nd Assessment—TUG-DT	(*n* = 26) 24.53 (±5.86)	(*n* = 16) 14.66 (±2.20)	<0.001

**Table 2 healthcare-13-03190-t002:** Comparative Results in TUG Test Assessment Pre- and Post-Intervention and Effect Size.

Group	TUG	Point Estimate	TUG DT	Point Estimate
With Suspected Cognitive Impairment	<0.001	−0.73 **	0.003	−0.59 *
No Suspected Cognitive Impairment	0.026	−0.56 *	0.44	−0.19

Legend: TUG (Timed Up and Go Test)*;* TUG-DT (Timed Up and Go Test Dual-task); Point Estimate (Effect Size); ** *p* < 0.01; * *p* < 0.05.

## Data Availability

The datasets presented in this article are not readily available because the data are part of an ongoing study. Requests to access the datasets should be directed to the corresponding author.

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
