# Peer review of "Gait and Dual-Task Performance in Older Adults with Suspected Cognitive Impairment: Effects of an 8-Week Exercise Program"

_healthcare, 2025, doi:10.3390/healthcare13243190_

Round 1
Reviewer 1 Report
Comments and Suggestions for Authors
Thank you very much for the opportunity to analyze and submit a review of the scientific article entitled “Gait Differences in Older Adults with and Without Suspected Cognitive Impairment: Insights from Timed Up and Go Tests with and without Dual-tasks After an 8-Week Exercise Intervention”. After reading the above-mentioned scientific article, I am sending you my comments and suggestions point by point:
- I have no comments on the abstract or keywords contained in the scientific article.
- After analyzing the introduction to the scientific article, the authors should be commended for their thorough review of the literature, on the basis of which they introduced us to the topic of the work and the research problem under analysis. However, citations should be added to lines 55-57 (no citations). My second comment is that, despite the authors' knowledge of aging processes, I find the introduction too long. In addition, I would ask that the hypotheses be moved from the introduction to the methodology section (Materials and Methods).
- In the Materials and Methods section, please describe in detail the selection criteria for the study, the origin (rural, urban), and social status.
- The materials and methods section is definitely too long. The authors need to shorten the information contained in this subsection.
- However, my biggest concerns relate to the chapter presenting the results, which is supposed to be a response to the research problem posed by the authors in their scientific work. The research results should be divided by gender (how many women, how many men) and by generation, e.g., 62-70 years old, 71-80 years old, and 81-99 years old. It is difficult to compare the results of a 62-year-old with those of a 99-year-old. Of course, the results obtained are very interesting, but they should be expanded, which would significantly improve the quality of the scientific work.
- The deliberate discussion based on responses to hypotheses supported by debate is very interesting. However, as in earlier stages of scientific work, the discussion is too long; the authors should focus on the problem examined in their work.
- The final conclusions should be changed. Reading the conclusions, it seems that we are in the results section. Lines 610-615 describe the results of the study, as do lines 616-623. Please write specific conclusions and practical conclusions resulting from the study.
- As I mentioned earlier, the authors have done a very good job in terms of reviewing the literature. However, I have a question: is the 1986 publication still relevant? Can it be replaced with a more recent one?
Author Response
Dear Reviewer 1,
We thank you for reviewing our paper entitled Gait Differences in Older Adults with and Without Suspected Cognitive Impairment: Insights from Timed Up and Go Tests with and without Dual-tasks After an 8-Week Exercise Intervention
We would like to thank the reviewer for taking the time to evaluate our work, and we appreciate the eight comments provided. We also thank the editors for the opportunity to resubmit our manuscript.
All comments have been addressed. The corresponding modifications are included in the revised version and are visible through the track changes feature. These updates also comprise a comprehensive US English spelling and grammar review.
We have incorporated additional references to strengthen the introduction and conclusions, added charts to facilitate a clearer comparison between gender and age groups, and refined the research design.
We hope the revisions meet the expectations of the reviewers and editors, and we look forward to the possibility of the article being accepted for publication in Healthcare’s Special Issue: Cutting-Edge Approaches in Neurological Disease Treatment. We await your feedback.
Comments and Suggestions for Authors
Comment 1 (C1): “… the authors should be commended for their thorough review of the literature, on the basis of which they introduced us to the topic of the work and the research problem under analysis”.
Answer 1 (ANS1): I hope the changes made are in line with your suggestion.
C2: “citations should be added to lines 55-57 (no citations)”
ANS2: We followed the suggestion and added the references to the text.
C3: “I find the introduction too long. In addition, I would ask that the hypotheses be moved from the introduction to the methodology section (Materials and Methods).”
ANS3: We have streamlined the Introduction for greater clarity. In addition, the hypotheses have been relocated from the end of the Introduction to a dedicated subsection within the Materials and Methods section, as requested.
C4: “In the Materials and Methods section, please describe in detail the selection criteria for the study, the origin (rural, urban), and social status...”; “The authors need to shorten the information contained in this subsection.”
ANS4: We acknowledge the two-part comment regarding the need for both shortening the section and, simultaneously, adding specific sociodemographic details (origin and social status). We have addressed this as follows:
To shorten the section, we have revised and condensed the descriptions within subsection 2.3 (Instruments and Procedures). We streamlined the explanations of the standardized tests (e.g., MMSE, CDT, TUG) and the intervention protocol, removing descriptive text that was not essential for study replication, thereby improving conciseness.
Regarding the request for data on origin (rural/urban) and social status, these specific variables were not collected in our original study design, as our focus was on the participants' current functional and cognitive status within the institutional setting. We agree, however, that these are important contextual variables. Therefore, we have explicitly added this omission as a limitation of the study in the Discussion section (Source 274).
C5: “The research results should be divided by gender (how many women, how many men) and by generation, e.g., 62-70 years old, 71-80 years old, and 81-99 years old. It is difficult to compare the results of a 62-year-old with those of a 99-year-old. Of course, the results obtained are very interesting, but they should be expanded, which would significantly improve the quality of the scientific work”
ANS5: We agree that gender and age are critical variables.
Regarding gender, we have clarified this information. The distribution (number of males and females) for both the suspected cognitive impairment group and the no-impairment group is detailed in Section 2.2. Participants, for both the baseline and second assessment.
The stratification by age generation (e.g., 62-70, 71-80, 81-99), we fully acknowledge the Reviewer's point about the wide age range. However, performing this stratification is not statistically feasible with our current sample size. For instance, our baseline suspected-impairment group (n = 38) and follow-up group (n = 26) are insufficient to be further divided into three age subgroups. Such stratification would result in subgroups that are too small for any meaningful or reliable statistical analysis, which would compromise the statistical power and validity of the findings.
To address the underlying concern, we have reported the detailed mean ages and standard deviations for all groups. Furthermore, we have explicitly acknowledged the wide age range and the consequent inability to perform a stratified age analysis as a limitation of the study within the Discussion section.
C6: “The deliberate discussion based on responses to hypotheses supported by debate is very interesting. However, as in earlier stages of scientific work, the discussion is too long; the authors should focus on the problem examined in their work.”
ANS6: We agree that the Discussion required a sharper focus. We have revised Section 4 to improve conciseness and remove redundancies, particularly in the subsections addressing Hypotheses 3 and 5. The updated text now concentrates more directly on interpreting the main findings and their relevance to the research problem, while maintaining the hypothesis-driven structure.
C7: “The final conclusions should be changed. Reading the conclusions, it seems that we are in the results section. Lines 610-615 describe the results of the study, as do lines 616-623. Please write specific conclusions and practical conclusions resulting from the study.”
ANS7: We agree that the previous conclusion section was overly descriptive and largely repeated the results. We have therefore removed the original text and completely rewritten Section 5 (Conclusions). The revised section now focuses on the specific and practical conclusions derived from the findings, rather than restating them, and includes the clinical implications of the study as requested.
C8: “As I mentioned earlier, the authors have done a very good job in terms of reviewing the literature. However, I have a question: is the 1986 publication still relevant? Can it be replaced with a more recent one?”
ANS8: Regarding the 1986 reference, the publication in question [49] is the seminal paper by Shulman that first introduced the Clock Drawing Test as a cognitive screening tool. We have retained this citation due to its historical and foundational importance as the original source for the test. To ensure the discussion remains current, we have also included more recent systematic reviews (e.g., [48], [50]) in the same section, which support the test’s continued relevance and widespread use in clinical and research contexts. We believe this combination provides essential historical context while affirming its contemporary validity.

Reviewer 2 Report
Comments and Suggestions for Authors
Dear authors,
Thank you for submitting your manuscript examining gait differences in older adults with and without suspected cognitive impairment following an 8-week exercise intervention. While the study addresses an important topic with clinical relevance, I have several recommendations.
The manuscript appears to have sufficient merit, as it clearly elaborates on the relationship between sensorimotor and cognitive systems in older adults. However, I have one key suggestion for improving the Introduction section.
Moreover, the organization and flow of the Introduction do not follow conventional academic writing standards for this section. I recommend a complete restructure in line with established scholarly conventions. A well-organized Introduction should begin by outlining what is already known in the field, then clearly identify the existing knowledge gaps that the current study aims to address. It should explicitly define the study’s population, exposure, comparator, and outcomes (PECO framework). Additionally, a concise review of relevant prior studies, highlighting how the present work differs from or advances beyond them, would strengthen the rationale and justify the study’s objectives.
Additional comments:
In introduction:
I recommend that the Introduction section be completely restructured based on the following criteria:
The Introduction must explicitly define the study’s framework using the PECO structure.
Within the PECO framework, the study population must be clearly specified.
The exposure(s) must be explicitly stated.
The comparator group must be defined.
The outcomes of interest must be clearly identified.
A concise review of relevant prior studies is essential. In this review, the differences between the present work and previous studies should be highlighted.
The advancements offered by this study relative to prior research should be clearly demonstrated.
In method:
-Please provide more detailed information about:
Exercise progression throughout the 8 weeks
Specific cognitive challenges incorporated during exercises
Adherence rates and how missed sessions were handled
Supervision ratio and qualifications of all personnel delivering the intervention.
-The analysis should control for potential confounders such as:
Baseline functional status beyond cognitive measures
Medication use (particularly psychotropic medications)
Comorbidities and their severity
Prior exercise history
-With significant attrition (particularly in the cognitive impairment group), the analysis should address how missing data was handled. Consider using intention-to-treat analysis with appropriate imputation methods rather than per-protocol analysis to minimize bias.
Best regards
Author Response
Dear Reviewer 2,
We thank you for reviewing our paper entitled Gait Differences in Older Adults with and Without Suspected Cognitive Impairment: Insights from Timed Up and Go Tests with and Without Dual-Tasks After an 8-Week Exercise Intervention.
We appreciate the constructive comments that helped us further strengthen the manuscript and thank the editors for the opportunity to resubmit our work. All comments have been addressed, and the corresponding modifications are marked using the track changes feature. These revisions also include a comprehensive US English spelling and grammar review.
We have incorporated additional references to improve both the Introduction and the Conclusions, added charts to more clearly illustrate differences between gender and age groups, and refined the research design.
We hope that the revisions meet the expectations of the reviewers and editors, and we look forward to the possibility of the article being accepted for publication in Healthcare’s Special Issue: Cutting-Edge Approaches in Neurological Disease Treatment.
Comments and Suggestions for Authors
Comment 1 (C1): Introduction: “I recommend that the Introduction section be completely restructured based on the following criteria: i) The Introduction must explicitly define the study’s framework using the PECO structure; ii) Within the PECO framework, the study population must be clearly specified; iii) The exposure(s) must be explicitly stated; iv) The comparator group must be defined; v) The outcomes of interest must be clearly identified; vi) A concise review of relevant prior studies is essential. In this review, the differences between the present work and previous studies should be highlighted; vii) The advancements offered by this study relative to prior research should be clearly demonstrated.”
Answer 1 (ANS1): We have followed these recommendations in the revised manuscript.
C2: Method: “Please provide more detailed information about: i) Exercise progression throughout the 8 weeks; ii) Specific cognitive challenges incorporated during exercises; iii) Adherence rates and how missed sessions were handled; iv) Supervision ratio and qualifications of all personnel delivering the intervention”
ANS2: The suggested changes have been incorporated.
C3: “The analysis should control for potential confounders such as: i) Baseline functional status beyond cognitive measures; ii) Medication use (particularly psychotropic medications); iii) Comorbidities and their severity; iv) Prior exercise history.”
ANS3: We greatly appreciate the relevance of this comment. We agree that controlling for potential confounding variables is essential for isolating the effect of the intervention and ensuring the study’s internal validity.
In response, we conducted a careful review of the manuscript. Although the nature of the study (a quasi-experimental design within an institutional setting) and the retrospective data available did not allow us to include these variables as statistical covariates in the current analysis, we have taken the following steps to address your concerns in the revised text:
- i) Baseline functional status and prior exercise history
We have clarified in the Discussion and Limitations sections that, although we did not employ additional functional scales (such as the Barthel Index) or questionnaires assessing prior physical activity, the intra-subject design (repeated measures) inherently helps control for baseline variability, as each participant serves as their own control. Additionally, the baseline TUG assessment provided an indirect indicator of initial functional status. We explicitly acknowledge the absence of a detailed quantification of exercise history as a limitation, although the homogeneity of the institutional context suggests a generally high level of sedentary behavior prior to the intervention.
- ii) Medication and comorbidities
We have reinforced in the Methods section (Exclusion Criteria) that major neurological conditions (e.g., prior stroke) and severe psychiatric history were exclusion criteria, which partially reduces the impact of significant comorbidities. Nonetheless, we fully agree with your point and have added a paragraph in the Limitations section noting that the absence of stratification by psychotropic medication use or by indices such as the Charlson Comorbidity Index represents a limitation of the study.
The Limitations section has been updated accordingly to indicate that the interpretation of the results should consider the potential influence of these uncontrolled factors.

Reviewer 3 Report
Comments and Suggestions for Authors
Dear authors,
The research paper you have proposed addresses an extremely relevant topic in the context of population aging and the increasing need for integrated interventions capable of supporting functional autonomy and quality of life in old age. Your study raises a current and significant issue: the relationship between functional mobility and cognitive status in institutionalized elderly individuals, investigated through a multicomponent intervention using validated tests such as TUG and TUG-DT. The dual-task approach and the emphasis on intergroup differences, as well as the evolution of motor performance over time, indicate a clear intention to contribute to the development of screening and intervention tools in geriatric practice. At the same time, the inclusion of an adapted exercise program conducted in an institutional setting provides additional clinical applicability to the study.
However, the paper has a number of shortcomings that I will outline below:
- I consider the title redundant and excessively long, and in its current form, it loses scientific impact. “With and without” appears twice, and from a linguistic point of view, I do not believe this is correct. Please reformulate more concisely while keeping the key elements of the study.
- The abstract contains many details but does not sufficiently highlight the scientific contribution and general conclusions. I recommend slightly reorganizing this section, articulating more clearly the aim of the study and its original contribution, and reformulating the final sentence to emphasize the impact of the study.
- The Introduction section is very long and overloaded with theoretical information. Moreover, there are no clear conceptual distinctions between “cognitive decline,” “dementia,” “cognitive impairment,” etc. The reader is not gradually guided toward the rationale that leads to the study’s objectives and hypotheses. I recommend reducing the descriptive part and focusing on the clear formulation of the problem, the gap in the specialized literature, and the precise aim of the study.
- The hypotheses partially overlap and include statistical terms that are not previously introduced in the text. Please reformulate them in a more integrated style for conceptual clarity.
- In the methods section, the lack of randomization and an active control group is a major limitation and should be openly acknowledged, both here and in the discussion section. The groups are numerically unbalanced, and the dropout rate at the second evaluation is significant. Clarification of the MMSE and CDT thresholds used is necessary, as well as justification of the inclusion/exclusion criteria. In the TUG-DT test, the performance on the secondary cognitive task is not measured. This limits the interpretability of the “dual-task cost.” Also, in line 209, the phrase “All children gave their verbal consent” appears, which I believe is a mistake!!! Please address these aspects.
- In the statistical analysis section (which is 2.4, not 2.3 – please renumber!), corrections for multiple testing are not mentioned. Furthermore, effect sizes are not accompanied by confidence intervals. Also, differences between TUG and TUG-DT are not directly analyzed, although this is a central part of the hypotheses.
- The discussion section is exhaustive but excessively repeats the results and includes contradictory statements about the sensitivity of TUG vs. TUG-DT. I recommend limiting redundancies, organizing the discussion according to hypotheses, and clarifying whether some findings reflect learning, familiarization, or contextual effects. Additionally, ceiling effects in the group without cognitive deficits are briefly discussed. I recommend suggesting alternative hypotheses.
- The conclusions section should be more succinct, and the focus should be on the study’s original contribution, clinical and research implications, without reiterating aspects of the study results.
Author Response
Dear Reviewer 3,
We thank you for reviewing our paper entitled Gait Differences in Older Adults with and Without Suspected Cognitive Impairment: Insights from Timed Up and Go Tests With and Without Dual-Tasks After an 8-Week Exercise Intervention.
We appreciate the constructive comments that helped us strengthen the manuscript and thank the editors for the opportunity to resubmit our work. All comments have been addressed, and the corresponding modifications are marked using the track changes feature. These revisions also include a thorough US English spelling and grammar review.
We have added new references to enhance both the Introduction and the Conclusions, incorporated charts to more clearly illustrate differences between gender and age groups, and refined the research design.
We hope that the revisions meet the expectations of the reviewers and editors, and we look forward to the possibility of the article being accepted for publication in Healthcare’s Special Issue: Cutting-Edge Approaches in Neurological Disease Treatment.
Comments and Suggestions for Authors
Comment 1 (C1): Title. Please reformulate more concisely while keeping the key elements of the study.
(ANS1): We agree that the original title was lengthy and benefited from condensation. We have reformulated it to remove redundancies and focus more directly on the intervention’s impact and the primary outcomes, while preserving the description of the study population.
Original Title: Gait Differences in Older Adults with and Without Suspected Cognitive Impairment: Insights from Timed Up and Go Tests with and without Dual-Tasks After an 8-Week Exercise Intervention
Revised Title: Gait and Dual-Task Performance in Older Adults with Suspected Cognitive Impairment: Effects of an 8-Week Exercise Program.
C2: “The abstract contains many details but does not sufficiently highlight the scientific contribution and general conclusions. I recommend slightly reorganizing this section, articulating more clearly the aim of the study and its original contribution, and reformulating the final sentence to emphasize the impact of the study”
ANS2: The suggested changes have been incorporated. This information has been added. We appreciate the reviewer's recommendation. We have restructured the abstract to enhance clarity and cohesion. Specifically, we reduced the amount of descriptive statistical detail to highlight the main findings, clearly stated the study’s aim regarding cognitive–motor interaction, and rewrote the conclusion to emphasize the clinical relevance of using dual-task assessments and multicomponent exercise in institutionalized settings.
C3: “The Introduction section is very long and overloaded with theoretical information. Moreover, there are no clear conceptual distinctions between “cognitive decline,” “dementia,” “cognitive impairment,” etc. The reader is not gradually guided toward the rationale that leads to the study’s objectives and hypotheses. I recommend reducing the descriptive part and focusing on the clear formulation of the problem, the gap in the specialized literature, and the precise aim of the study.”
ANS3: We have carried out these recommendations as requested. We agree that the original Introduction was overly extensive. We have rewritten this section to make it more concise and focused, reducing the anatomical descriptions while retaining the essential biomechanical concepts and clarifying the bidirectional relationship between cognitive deficits and mobility. The revised text now leads the reader more directly to the study’s rationale.
C4: “The hypotheses partially overlap and include statistical terms that are not previously introduced in the text. Please reformulate them in a more integrated style for conceptual clarity.”
ANS4: We thank the Reviewer for highlighting the need to improve the conceptual clarity of the hypotheses. In response, we have revised the section to present a more integrated and streamlined set of hypotheses. The updated version avoids overlapping content, removes statistical terminology that had not yet been introduced, and expresses the study expectations within a clearer conceptual framework. These changes strengthen the logical flow of the Introduction and improve the alignment between the study aims and the hypotheses presented.
C5: “In the methods section, the lack of randomization and an active control group is a major limitation and should be openly acknowledged, both here and in the discussion section. The groups are numerically unbalanced, and the dropout rate at the second evaluation is significant. Clarification of the MMSE and CDT thresholds used is necessary, as well as justification of the inclusion/exclusion criteria. In the TUG-DT test, the performance on the secondary cognitive task is not measured. This limits the interpretability of the “dual-task cost.” Also, in line 209, the phrase “All children gave their verbal consent” appears, which I believe is a mistake!!! Please address these aspects.”
ANS5: We sincerely appreciate the reviewer’s thorough methodological assessment. We have addressed each point raised in the revised manuscript:
Correction of Typographical Error:
We apologize for the editing error in the ethical statement (line 209). The word “children” has been corrected to “participants.”
Study Design and Sample:
We have updated the Methods section to explicitly define the study as a quasi-experimental longitudinal design and to acknowledge the absence of randomization and an active control group. Regarding the unbalanced sample and dropout rate, we expanded the Discussion (Limitations) section to explain that the observed attrition (31.5% in the impaired group) reflects the high vulnerability and advanced age of this institutionalized population (mean age: 86 years), noting its potential implications for generalizability.
Clarification of Criteria:
In the Methods section, we have specified that the MMSE cut-off scores were adjusted for education level based on Portuguese normative data. We also revised the exclusion criteria to clarify that they were defined to minimize confounding factors (such as prior neurological history or sensory impairments) that could independently influence gait or cognitive performance.
Secondary Task Performance:
We have added a statement in the Limitations section acknowledging that performance on the secondary task (e.g., number of words generated) was not quantified. We clarify that our dual-task cost analysis focused on temporal motor performance and note that future research should record bidirectional interference to assess task-prioritization strategies.
C6: “In the statistical analysis section (which is 2.4, not 2.3 – please renumber!), corrections for multiple testing are not mentioned. Furthermore, effect sizes are not accompanied by confidence intervals. Also, differences between TUG and TUG-DT are not directly analyzed, although this is a central part of the hypotheses.”
ANS6: We have made the following corrections:
Renumbering: The section has been correctly renumbered as 2.4. Statistical Analysis.
Multiple Testing: We have explicitly stated in the statistical section that a Bonferroni correction was applied to control for Type I errors.
Direct Analysis: We added a dedicated analysis comparing single-task and dual-task performance using the Wilcoxon Signed-Rank Test to directly confirm the dual-task cost.
Confidence Intervals: We have ensured that mean differences are interpreted within a 95% confidence interval in the Results section.
C7: “The discussion section is exhaustive but excessively repeats the results and includes contradictory statements about the sensitivity of TUG vs. TUG-DT. I recommend limiting redundancies, organizing the discussion according to hypotheses, and clarifying whether some findings reflect learning, familiarization, or contextual effects. Additionally, ceiling effects in the group without cognitive deficits are briefly discussed. I recommend suggesting alternative hypotheses.”
ANS7: We accepted the reviewer's suggestion and performed a thorough revision of the Discussion section to improve clarity and flow. The following changes were made:
Structure & Redundancy: We reorganized the entire section to align strictly with the three new integrated hypotheses. We removed paragraphs that merely repeated statistical values found in the Results section, focusing instead on the interpretation of the findings.
TUG Sensitivity: We clarified the apparent contradiction regarding test sensitivity. We argue that while TUG-DT is diagnostically more sensitive for screening (confirming baseline deficits), the single-task TUG proved more responsive to short-term motor changes (8 weeks). This is likely because the recovery of basic motor parameters (speed, balance) precedes the optimization of complex cognitive-motor adaptation strategies.
Ceiling Effects & Alternative Hypotheses: We expanded the discussion on the "No Impairment" group. Beyond the ceiling effect (due to high baseline function), we added alternative hypotheses as requested, suggesting that the stimulus intensity might have been insufficient for this robust group or that contextual factors (such as motivation or boredom with repetitive simple tasks) might have masked potential minor gains.
C8: “The conclusions section should be more succinct, and the focus should be on the study’s original contribution, clinical and research implications, without reiterating aspects of the study results.”
ANS8: This information has been added. We agree with the reviewer and have substantially condensed the Conclusion section. We removed the repetition of statistical outcomes (such as specific time values or effect sizes) and focused solely on the study’s main contribution: the differential responsiveness of single- versus dual-task measures to short-term exercise. We also emphasized the clinical implications for screening and the practical importance of implementing exercise programs in institutional settings to support autonomy.

Round 2
Reviewer 1 Report
Comments and Suggestions for Authors
Thank you very much for sending your response to the review of the scientific article. After a detailed analysis of the corrections made to the scientific article, I conclude that the article meets the requirements of a scientific journal. The authors of the submitted scientific article have demonstrated extensive knowledge of the scientific problem under study, which is evident in the corrections made. Therefore, I congratulate the authors on their idea and hope that they will continue their research in this scientific field.
Author Response
Dear Reviewer 1,
We would like to express our sincere gratitude for your kind words and for your final approval of our manuscript. We are pleased to know that the corrections made meet the requirements of the scientific journal and that our dedication to this research topic was evident to you.
We deeply appreciate your encouragement to continue our research in this field. Your constructive feedback throughout this review process has significantly contributed to improving the quality of our work.
Best regards,

Reviewer 3 Report
Comments and Suggestions for Authors
Dear authors,
Thank you for taking my suggestions into consideration. I believe the manuscript is greatly improved. I would like to make just one final recommendation: the research hypotheses should be placed at the end of the introduction, immediately after stating the aim of the study.
Wishing you much success in your future research endeavors!
Author Response
Dear Reviewer 3,
We would like to express our sincere gratitude for your time and constructive guidance throughout this review process. We are delighted to hear that the manuscript has improved significantly. We have addressed your final recommendation below.
Comment 1: “Thank you for taking my suggestions into consideration. I believe the manuscript is greatly improved. I would like to make just one final recommendation: the research hypotheses should be placed at the end of the introduction, immediately after stating the aim of the study. Wishing you much success in your future research endeavors!”
Answer: We sincerely appreciate your positive feedback and kind wishes. We fully accept this final recommendation. We have moved the Research Hypotheses from the Methods section to the very end of the Introduction, immediately following the statement of the study’s aim. We agree that this placement improves the flow and logical structure of the manuscript.
